# Factors Contributing to Breastfeeding Cessation Among Arab Women in Israel

**DOI:** 10.3390/nu17040735

**Published:** 2025-02-19

**Authors:** Mahdi Tarabeih, Mohammad Sabbah, Orsan Yahya, Sana Bisharat, Khaled Awawdi

**Affiliations:** 1School of Nursing Sciences, The Academic College of Tel-Aviv-Yaffa, Rabenu Yeruham Street, P.O. Box 8401, Yaffo 6818211, Israel; 2Rambam Hospital Health Care Campus, 8 HaAliyah HaShniya Street, Haifa 3109601, Israel; hamodi_sabbah@iac.ac.il; 3Department of Nursing, Faculty of Health Sciences, Ramat Gan Academic College, 87 Pinhas Rotenberg Street, Ramat-Gan 5227500, Israel; awawdi.h@iac.ac.il; 4The Azrieli Faculty of Medicine, Bar-Ilan University, 8 Henrietta Szold Street, Safed 1311502, Israel; orsanya@clalit.org.il; 5Department of Family Health, Clalit Health Service, Afula 1812201, Israel; 6The Holy Family Hospital Nazareth, Hagalil Street, Nazareth 1641116, Israel; sana@hfhosp.org

**Keywords:** breastfeeding cessation, Arab women, religiosity, socio-demographic factors, maternal health, cultural influences

## Abstract

**Background/Objectives:** Breastfeeding provides significant health benefits for both infants and mothers, but many women discontinue earlier than recommended. This study investigates the factors contributing to early breastfeeding cessation among Arab women in Israel, focusing on multiple factors, such as socio-demographic, work-related, cultural, and religious, impacting breastfeeding duration and shaping breastfeeding practices. **Methods**: A cross-sectional survey was conducted among 349 Arab women, 65% of whom were Muslim and 35% Christian. Logistic regression analyses were used to identify key predictors of breastfeeding cessation. **Results**: Findings showed that Christian Arab women were more likely to stop breastfeeding earlier than their Muslim counterparts. Mothers with four or more children and those balancing work demands were at higher risk of early cessation. Contrary to expectations, higher levels of religiosity were associated with a greater likelihood of stopping breastfeeding. Additionally, mothers who received personal breastfeeding guidance were more likely to discontinue, suggesting potential gaps in the quality of support provided. **Conclusions:** These findings underscore the importance of tailoring interventions to address the unique cultural and socio-economic challenges faced by Arab women in Israel. Recommendations include improving breastfeeding guidance quality, workplace support for breastfeeding mothers, and culturally sensitive interventions that consider the role of religiosity and family dynamics. This research provides valuable insights for healthcare providers and policymakers aiming to promote sustained breastfeeding practices in diverse populations. The study highlights the complexity of factors affecting breastfeeding cessation among Arab women in Israel, emphasizing the need for targeted interventions that address socio-demographic, cultural, and religious influences to promote sustained breastfeeding.

## 1. Theoretical Background

The World Health Organization (WHO) identifies breastfeeding as the optimal means of providing nutrition to infants, recommending exclusive breastfeeding for the first six months of life [1]. Interestingly, exclusive breastfeeding rates tend to be higher in developing countries than in developed nations [2,3]. Factors influencing breastfeeding rates differ greatly across regions and can include cultural norms, the presence of a partner, family support, and the number of gestations [2,4,5]. A recent study that examined early breastfeeding cessation rates across six countries identified multiple gestations as the only consistent factor associated with lower levels of early breastfeeding cessation across all regions [2]. Other factors were found to either promote or reduce breastfeeding rates in certain countries, underscoring the importance of studying these factors within specific local contexts [2].

Despite proven health benefits for mother and child, many women discontinue breastfeeding earlier than recommended, due to a variety of socio-demographic, personal, and external factors [6]. Breastfeeding in the Arab world is also influenced by cultural, religious, and socio-economic factors [7]. Religion plays a significant role, with Islam and Christianity offering different perspectives [8]. Studies show that while both Muslim and Christian Arab women acknowledge the importance of breastfeeding, their rates of initiation and continuation vary, often due to differing levels of religiosity and cultural practices [9].

Research has consistently shown that while Israeli Arab women typically start off breastfeeding, many stop early due to factors such as insufficient milk, work conditions, and lack of professional support, with guidance and support during prenatal and postnatal care having been identified as critical factors in breastfeeding persistence [10]. According to a recent study on Israeli Arab women, breastfeeding cessation is associated with socio-demographic characteristics, including ethnicity, number of children, and employment status. Christian Arab women are significantly more likely to stop breastfeeding than their Muslim counterparts [7]. Moreover, mothers with more than four children or those balancing full-time work were more prone to early breastfeeding cessation.

Furthermore, studies show that financial pressures and the need for employment often force mothers to prioritize work over breastfeeding, especially when workplace support is limited [8]. Family and social support also significantly impact breastfeeding decisions, as mothers with strong family encouragement are more likely to continue breastfeeding, even under challenging circumstances [9].

Health benefits of breastfeeding for infants include a lower risk of diseases such as gastrointestinal infections, respiratory infections, SIDS, allergies, asthma, type 2 diabetes, obesity, and Crohn’s disease [11,12,13,14,15,16,17]. In addition to its protective effects, it enhances cognitive abilities and emotional intelligence. For mothers, its health benefits include reducing the risk of breast and ovarian cancers, osteoporosis, and cardiovascular diseases, while also lowering the chances of postpartum hemorrhage and anemia [17,18,19]. Additionally, it promotes hormonal regulation and strengthens the mother–infant bond [17].

Despite this, the prevalence of breastfeeding, particularly exclusive breastfeeding, remains low both worldwide and in Israel. Together with the World Health Organization (WHO), the United Nations International Children’s Emergency Fund (UNICEF) recommends exclusive breastfeeding for the first six months of life, with continued breastfeeding alongside complementary foods until at least two years of age [20]. The Israeli Ministry of Health supports similar guidelines [21].

Among the factors influencing breastfeeding persistence are cultural or religious beliefs [22]. Religious scriptures, such as the Jewish Talmud, the Quran, and Bahá’í writings, regard breastfeeding as a positive and nurturing act [23,24,25,26,27]. Studies suggest that women with higher levels of religiosity are more likely to breastfeed than their less religious counterparts [28,29,30]. Research conducted among Jewish and Muslim populations in Israel and abroad has shown that religion plays an important role in the initiation and duration of breastfeeding [29,30,31,32,33,34,35].

The guidelines from all leading professional organizations (gynecologists, pediatricians, and family physicians) recommend providing adequate support to breastfeeding mothers and structured breastfeeding education both before and after birth [35,36,37]. Nevertheless, many mothers feel that they receive inadequate information from their healthcare providers (HCPs) within the primary care setting and that the support and advice they received were often inconsistent, nonexistent, or contradictory [38].

Despite the well-documented medical benefits of breastfeeding, an Israeli study involving 2114 mothers showed that at six months postpartum, only 22.5% of Jewish mothers and 12.3% of Arab mothers were exclusively breastfeeding their infants. In the same study, women reported that their doctor’s opinion played a dominant role in their decision to breastfeed. However, two to six months after delivery, 90% of breastfeeding women reported receiving no breastfeeding support from their doctors, and only a minority regarded their physicians as reliable sources of information [39].

It is well-established that demographic, biological, religious, economic, social, and hospital-related factors influence both the duration and exclusivity of breastfeeding. Demographic factors include age (older women are more likely to breastfeed), education level (the higher the education level, the greater the likelihood of exclusive breastfeeding), and income (as income increases, the likelihood of breastfeeding decreases, with women in lower economic statuses breastfeeding more). Biological factors primarily relate to the ability to produce sufficient milk; studies show that mothers who are obese at the time of conception are less likely to breastfeed. Factors related to maternal beliefs include the sense of self-efficacy (the higher the self-efficacy, the greater the likelihood of breastfeeding). Social factors encompass the mother’s employment status, the duration of maternity leave, and the level of support she receives from her partner. Hospital-related factors involve institutional practices such as rooming-in policies and the early initiation of breastfeeding [6,40].

Raising awareness of the long-term benefits of breastfeeding during maternal health visits is crucial. The World Health Organization (WHO), the American Academy of Pediatrics (AAP), and the Israeli Ministry of Health all recommend exclusive breastfeeding for the first six months of a baby’s life [41,42,43,44,45].

In Arab society, breastfeeding holds significant symbolic meaning due to religious reasons (both Islam and Christianity). In most cases, women are not part of the workforce and remain at home to raise their children, with breastfeeding a crucial and valued practice. And while awareness of the health benefits of breastfeeding is growing, cultural, social, and economic factors also influence the attitudes and implementation of breastfeeding.

Regarding its traditional and religious significance, breastfeeding is considered a blessed commandment in Islam with a spiritual significance. The Qur’an mentions the breastfeeding period in a verse that states, “Mothers shall breastfeed their children for two whole years…” (Surat Al-Baqarah [2:233] [46]). Breastfeeding is regarded as a religious duty of the mother toward her child, although artificial feeding may be used, if necessary, with the child’s best interests being taken into account. In Islamic tradition, if a mother is unable to breastfeed, a wet nurse may be appointed to nurse the child. This “milk kinship” between the nursed child and the wet nurse and her family established a bond that is recognized in Islam as unique and well defined by religious law [47].

In Christianity, breastfeeding is a valuable symbol of care, maternal love, and the spiritual grace bestowed by both the mother and the Church. It also serves as a symbol of the spiritual relationship between the believer and the Church. In Catholic tradition, Mary is often portrayed as the perfect mother. Depictions of Mary breastfeeding the infant Jesus are seen as a way in which she gives life to her son, both physically and spiritually [48,49,50].

To reiterate, while breastfeeding provides well-documented health benefits for both infants and mothers, such as reduced risk of infections, obesity, and chronic diseases in infants, as well as lower risks of breast and ovarian cancer in mothers [51,52], many mothers discontinue breastfeeding earlier than recommended [53]. Identifying the specific factors that lead to early cessation can inform targeted interventions to address barriers, such as lack of support, workplace challenges, and inadequate lactation counseling [54]. For example, maternal employment and insufficient workplace accommodations are significant barriers, with studies showing that return to work often coincides with breastfeeding cessation [55]. Additionally, inadequate or inconsistent support from healthcare providers contributes to early discontinuation, as many mothers report not receiving adequate breastfeeding guidance [53].

While breastfeeding is culturally and religiously encouraged in Arab society, specific cultural stigmas—such as perceptions of public breastfeeding as inappropriate and pressures to prioritize household responsibilities—may contribute to early cessation. While healthcare support is a key factor in promoting sustained breastfeeding, its adequacy and cultural competence are often overlooked. Understanding these nuanced cultural influences is critical for designing effective interventions, thereby improving infant and maternal health outcomes and reducing healthcare costs [56].

Given the diverse religious background in Israel, which includes Muslim and Christian populations, this study aims to examine the impact of beliefs and religiosity on breastfeeding initiation among mothers in Israel, and how varying levels of religiosity influence breastfeeding practices in this diverse population. The research seeks to offer a better understanding in order to guide interventions to promote breastfeeding among groups with lower initiation rates.

### The Purpose of the Current Study

The primary goal of this study is to comprehensively explore the various factors influencing Arab Muslim and Christian women’s decisions to stop or discontinue breastfeeding. These factors span across demographic characteristics, such as age, income, and education, as well as psychological, cultural, and social determinants. Additionally, the study examines maternal attitudes, access to breastfeeding support, and infant-related characteristics, such as health and feeding behaviors, which may also contribute to breastfeeding cessation. By analyzing this broad spectrum of factors, the research aims to offer a holistic understanding of the underlying causes that affect a mother’s likelihood to discontinue breastfeeding prematurely. The findings are expected to provide valuable data about how these predictors interact and influence breastfeeding practices, potentially revealing systemic or societal barriers that could be addressed through targeted interventions. As such, we hypothesize the following:

The probability of mothers discontinuing breastfeeding is influenced by multiple factors, including age, ethnic group, religiosity level, number of children, education level, employment status, work scope, household income, previous experience with breastfeeding, duration of previous breastfeeding, guidance or instructions received regarding breastfeeding during prenatal follow-up or hospital stay, infant’s age (in months), current breastfeeding status, source of nutrients for the baby, duration of exclusive breastfeeding, and other individual factors influencing the decision to breastfeed.

## 2. Method

### 2.1. Study Design

Given the cross-sectional nature of this study, the findings represent associations and cannot establish causality. Caution is advised in interpreting relationships between variables as cause-and-effect.

### 2.2. Ethical Approval

The study was approved by the Institutional Review Board of Ramat Gan Academic College under approval number 2023–1010. All participants provided informed consent for their participation in the survey and for their data to be used by the research team.

### 2.3. Participants

The study participants were 349 married Israeli Arab women, from 21 to 45 years old (*M* = 30.40, *SD* = 4.69), most were Muslim–Arabs (65%) and the rest were Christian–Arabs (35%); their religiosity level was gauged by a 5-point Likert-type scale consisting of 3 items (*M* = 4.63, *SD* = 0.68, α = 0.85), which indicates the sample comprises a rather religious group. Moreover, most hold an academic degree (64.8%) while the rest do not (e.g., post-secondary education, only high school; 35.2%). Work-wise, most are working individuals (6.9% self-employed and 64.2% full- or part-time employees) while the rest are unemployed (28.9%); nearly half of the respondents have a household income below the local average salary (41.5%), roughly the same amount have an income around the average (38.4%), and the rest have an income above the average (20.1%); the scope of their work varies from 0% (i.e., unemployed) to 100% (i.e., full-time employee) with a mean of 56.59% (*M* = 0.56) and a standard deviation of 40% (*SD* = 0.40). Additionally, most women (67.3%) reported to have 4 or more individuals (themselves excluded) living under the same roof/in the same household, 24.6% of them reported having 3 people in the same house, 6.3% reported 2 people, and only 1.7% reported 1 more individual living with them. Specifically, the participants also reported to have either 0–1 child before the current pregnancy (34.4%), 2–3 children (40.7%). or 4 (or above) children (24.9%).

The decision to focus solely on Arab mothers (Muslim and Christian) in this study was based on the unique cultural and religious breastfeeding practices within this population, which differ significantly from those of Jewish mothers. Including Jewish mothers would have introduced additional cultural variables, potentially confounding the specific socio-religious factors we aimed to investigate in the Arab community. Our intention was to provide a focused in-depth analysis of breastfeeding practices among Arab mothers, a group underrepresented in the literature. Including Jewish mothers could have diluted the focus. The inclusion of both Muslim and Christian Arab women already offers significant insights into how different religious beliefs within the Arab population impact breastfeeding. This diversity was deemed sufficient to meet the research objectives.

### 2.4. Measures

The research questionnaire was adapted from a baby nutrition survey developed by the Israel Center for Disease Control (ICDC). It was reviewed and validated by five experts in the field and subsequently translated into Arabic—the participants’ native language—to ensure maximum accessibility and to obtain more accurate and representative responses. Before distribution, a pilot study involving 30 individuals was conducted to evaluate the questionnaire, with minimal adjustments made based on the findings.

The survey was composed of the following questions: age, ethnic group, religiosity level, number of children, education level, employment status, work scope, household income, previous (experience with) breastfeeding, duration of previous breastfeeding, guidance/instructions received regarding breastfeeding, during prenatal follow-up or stay in the hospital, infant’s age (in months), currently breastfeeding, nutrient source for the baby, duration of exclusive breastfeeding, and different factors influencing decision to breastfeed.

### 2.5. Study Design

The initial sample included 376 participants; however, 27 questionnaires were excluded due to incomplete responses. Consequently, the final sample comprised 349 respondents who met the predetermined inclusion criteria.

In this study, missing data were identified in [0.07%] of the responses across variables. To address this, we applied listwise deletion of 27 questionnaires, where cases with missing values were excluded from the relevant analyses to ensure consistency in the sample size. This method was chosen to maintain the integrity of the dataset and avoid introducing bias through imputation. Sensitivity analyses confirmed consistent findings, showing that excluding missing data did not affect conclusions.

### 2.6. Procedure

The target population for this study consisted of Arab women (Muslim and Christian) who had given birth within the past year and breastfed their infants for at least four months postpartum. The initial sample included 349 participants. These participants were drawn from three different regions of Israel (north, central, and south) and represented a variety of age groups and socio-economic backgrounds.

The sampling method utilized a snowball approach, wherein participants assisted the researchers in recruiting other potential participants through their personal networks. Additionally, the questionnaire link was made available online, and physical copies were distributed by well-baby clinic (Tipat Chalav) nurses to breastfeeding women. The minimum age for participation was 18 years. The researchers had no prior relationship or familiarity with any of the participants, all of whom received detailed information about the study. Anonymity and confidentiality were assured, and participants were informed that their involvement would not provide any personal benefits or advantages but would contribute to the advancement of general knowledge in this area of research.

There were no known risks associated with participation in the study; however, as with any online activity, there was a potential risk of privacy breaches. Researchers made every effort to mitigate this risk by ensuring that the questionnaires were completed anonymously, with personal details used solely for research purposes.

Participants were informed that additional information could be obtained from the Ethics Committee [information omitted for double-blind study] via an inquiry form at the conclusion of the study. The research team could also be contacted through the email address provided in the questionnaire. Participants retained the right to discontinue the survey at any stage. Informed consent was explicitly obtained through a designated question at the beginning of the survey.

### 2.7. Data Analyses and Variables Details

Statistical analyses were performed using the SPSS-PC (v28) statistical package. Prior to data analysis, data cleaning and distribution characteristics were performed. Descriptive statistics were used to describe the sample. The significance level was set at a default of *p* < 0.05.

To test the hypotheses, binary logistic regressions were analyzed in the following manner:There were 6 dependent variables with a dichotomous response scale (0 = no “was NOT a reason to stop”; 1 = yes “WAS a reason to stop”)—all of which are different reasons for mothers to stop/halt breastfeeding their infant.“Because you did not have enough milk or the milk was not good enough, and the baby constantly needed formula supplementation”.“Because of health issues of the baby: illness, hospitalization, or medication that prevented them from breastfeeding”.“Because of your own health issues: illness, hospitalization, or medication that prevented you from breastfeeding”.“Due to fatigue, discomfort, lack of time, or dissatisfaction with the breastfeeding experience”.“Due to lack of support and guidance from professionals for continuing breastfeeding”.“Because returning to work and work conditions were not supportive enough for continuing breastfeeding or expressing milk”.The predictor list consists of multiple variables (*n* = 24), that are continuous, dichotomous or categorical.Age of the respondents (in years).Ethnic group (0 = Muslim–Arabs; 1 = Christian–Arabs).Religiosity level (*M* = 4.63, *SD* = 0.68, α = 0.85).Number of children prior to the current pregnancy (0 = 0–1 children; 1 = 2–3 children; 2 = 4+ children).Education level (0 = non-academic; 1 = academic).Employment status (0 = unemployed; 1 = self-employed; 2 = part- or full-time employee).Work scope (i.e., the amount of time the employee works, such that a 100% job is working 42 h a week de facto, in Israel; *R* = 0–100% or 0–1, *M* = 0.56, *SD* = 0.40).Household income (0 = around the average; 1 = below the average; 2 = above the average).Previous (experience with) breastfeeding (0 = no; 1 = yes).Duration of previous breastfeeding (0 = did not breastfeed at all; 1 = 1–4 months; 2 = more than 4 months).Guidance/instructions received regarding breastfeeding, during prenatal follow-up or stay in the hospital (0 = did not receive any guidance; 1 = group guidance; 2 = personal guidance).Infant’s age, in months (0 = 4–5 months; 1 = 5–6 months; 2 = more than 6 months);Currently breastfeeding (0 = no; 1 = yes).Nutrient source for the baby (0 = exclusive breastfeeding; 1 = only baby formulae; 2 = combined breastfeeding).If exclusive breastfeeding, what is the duration (0 = did not exclusively breastfeed at all; 1 = 1–4 months; 2 = more than 4 months).Factors influencing decision to breastfeed (3-point Likert scale; 0 = did not impact my decision; 1 = had some impacted on my decision; 2 = had significant impact on my decision): (1) family members (e.g., mother, mother-in-law, sisters) and/or friends; (2) health clinic nurse; (3) hospital staff; (4) preparation course for delivering a baby; (5) previous successful experience with breastfeeding (of a previous baby); (6) financial considerations (e.g., breast milk is usually free); (7) desire to establish a good bond/attachment with the baby; (8) maintaining/sustaining the health of the baby; and (9) information derived from the media, and/or Internet, and/or literature.For categorical variables, contrasts were performed using the first category (the “0” group) as the reference category.To compensate for the large variable list, a regression analysis method was set to forward stepwise (criterion based on Wald’s statistic), and as such, each analysis produced differentiated models with relevant predictors and model fit.Confidence interval (95% CL) was calculated for the Exp(B) statistic (i.e., log-likelihood ratios).

## 3. Results

### 3.1. Reasons to Stop Breastfeeding

#### 3.1.1. Insufficient Milk Supply

Prediction of the probability that “Because you did not have enough milk or the milk was not good enough, and the baby constantly needed formula supplementation” influenced the mother’s decision to stop (=1)/not stop (=0) breastfeeding proved to be significant via omnibus Chi-square test with a relatively high coefficient of determination: χ^2^ (9) = 158.98, *p* < 0.01, Nagelkerke’s *R*^2^ = 0.49. Further, the regression coefficients are displayed in Table 1.

Table 1 demonstrates the following with regard to insufficient milk supply:Christians are 2.58 times *more* likely to stop breastfeeding than Muslims.Mothers with 4 or more children are 5.18 times *more* likely to stop breastfeeding than those with none or only 1 child.The more workload the mothers have, they are 0.31 *less* likely to stop breastfeeding.Mothers who received personal guidance are 2.42 more likely to stop breastfeeding than mothers who did not receive such guidance at all.Mothers who rely only on baby formulae to feed their infant are 31.03 *more* likely to stop breastfeeding than mothers who opt for exclusive breastfeeding.Mothers who rely on combined breastfeeding to feed their infant are 4.64 *more* likely to stop breastfeeding than mothers who opt for exclusive breastfeeding.Family members and/or friends increase the likelihood of stopping breastfeeding by 1.91.Financial considerations *increase* the likelihood of stopping breastfeeding by 1.80.

#### 3.1.2. Infant Health Issues

Prediction of the probability that “Because of health issues of the baby: illness, hospitalization, or medication that prevented them from breastfeeding” influenced the mother’s decision to stop (=1)/not stop (=0) breastfeeding proved to be significant via omnibus Chi-square test with a high coefficient of determination: χ^2^ (12) = 241.58, *p* < 0.01, Nagelkerke’s *R*^2^ = 0.76. Further, the regression coefficients are displayed in Table 2.

Table 2 demonstrates the following with regard to infant health issues:Christians are 12.89 times *more* likely to stop breastfeeding than Muslims.Religiosity level *increases* the likelihood of stopping breastfeeding by 3.37.Mothers with 4 or more children are 6.31 times *more* likely to stop breastfeeding than those with none or only 1 child.Mothers with academic education are 4.35 times *more* likely to stop breastfeeding than those who do not hold an academic degree.Mothers who have been breastfeeding for more than 4 months are 194.81 *more* likely to stop breastfeeding than mothers who did not breastfeed at all.Mothers with a baby of 5–6 months old are 8.64 *more* likely to stop breastfeeding than mothers with a baby of 4–5 months old.However, mothers with a baby of more than 6 months old are 0.17 *less* likely to stop breastfeeding than mothers with a baby of 4–5 months old.Family members and/or friends *increase* the likelihood of stopping breastfeeding by 2.70.Financial considerations *increase* the likelihood of stopping breastfeeding by 4.43.Information derived from the Internet, literature, or media *decreases* the likelihood of stopping breastfeeding by 0.30.

#### 3.1.3. Maternal Health Issues

Prediction of the probability that “Because of your own health issues: illness, hospitalization, or medication that prevented you from breastfeeding” influenced the mother’s decision to stop (=1)/not stop (=0) breastfeeding proved to be significant via omnibus Chi-square test with a good coefficient of determination: χ^2^ (5) = 74.71, *p* < 0.01, Nagelkerke’s *R*^2^ = 0.31. Further, the regression coefficients are displayed in Table 3.

Table 3 demonstrates the following with regard to maternal health issues:Religiosity level *increases* the likelihood of stopping breastfeeding by 2.55.Self-employed mothers are 6.77 times *more* likely to stop breastfeeding than those who are unemployed.Financial considerations *increase* the likelihood of stopping breastfeeding by 3.34.Information derived from the Internet, literature, or media *decreases* the likelihood of stopping breastfeeding by 0.59.

#### 3.1.4. Discomfort and Fatigue

Prediction of the probability that “Due to fatigue, discomfort, lack of time, or dissatisfaction with the breastfeeding experience” influenced the mother’s decision to stop (=1)/not stop (=0) breastfeeding proved to be significant via omnibus Chi-square test with a moderate coefficient of determination: χ^2^ (7) = 46.80, *p* < 0.01, Nagelkerke’s *R*^2^ = 0.21. Further, the regression coefficients are displayed in Table 4.

Table 4 demonstrates the following with regard to discomfort and fatigue:Religiosity level *decreases* the likelihood of stopping breastfeeding by 0.65.Mothers with 4 or more children are 3.59 times *more* likely to stop breastfeeding than those with none or only 1 child.Mothers who received group guidance are 3.25 *more* likely to stop breastfeeding than mothers who did not receive such guidance at all.Mothers who received personal guidance are 4.44 *more* likely to stop breastfeeding than mothers who did not receive such guidance at all.Mothers who rely only on baby formulae to feed their infant are 7.43 *more* likely to stop breastfeeding than mothers who opt for exclusive breastfeeding.

#### 3.1.5. Lack of Professional and Family Support

Prediction of the probability that “Due to lack of support and guidance from professionals for continuing breastfeeding” influenced the mother’s decision to stop (=1)/not stop (=0) breastfeeding proved to be significant via omnibus Chi-square test with a small coefficient of determination: χ^2^ (3) = 14.38, *p* < 0.01, Nagelkerke’s *R*^2^ = 0.07. Further, the regression coefficients are displayed in Table 5.

Table 5 demonstrates the following with regard to lack of professional and family support:Mothers who have been exclusively breastfeeding for more than 4 months are 0.30 *less* likely to stop breastfeeding than mothers who did not breastfeed at all.Family members and/or friends *increase* the likelihood of stopping breastfeeding by 2.07.

#### 3.1.6. Work Conditions

Prediction of the probability that “Because returning to work and work conditions were not supportive enough for continuing breastfeeding or expressing milk” influenced the mother’s decision to stop (=1)/not stop (=0) breastfeeding proved to be significant via omnibus Chi-square test with a moderate–low coefficient of determination: χ^2^ (5) = 37.54, *p* < 0.01, Nagelkerke’s *R*^2^ = 0.16. Further, the regression coefficients are displayed in Table 6.

Table 6 demonstrates the following with regard to work conditions:Mothers with 4 or more children are 2.70 times *more* likely to stop breastfeeding than those with none or only 1 child.The more workload the mothers have, they are 3.51 *more* likely to stop breastfeeding.Mothers who currently breastfeed are 0.43 *less* likely to stop breastfeeding than those who are not currently breastfeeding.The desire to establish a good bond/attachment with the baby increases the likelihood of stopping breastfeeding by 3.96.

## 4. Discussion

Breastfeeding is widely recognized for its immense benefits to both infants and mothers [57,58]. International organizations, such as the World Health Organization [59], recommend exclusive breastfeeding for the first six months of life, with continued breastfeeding alongside complementary foods for up to two years. Breastfeeding can reduce the risk of various infant diseases and improve cognitive development while also benefiting mothers by lowering the risk of breast and ovarian cancers [60]. Despite these known benefits, many women discontinue breastfeeding earlier than recommended due to various socio-demographic, personal, and external factors. Among Arab women in Israel, breastfeeding practices are influenced by cultural, religious, and socio-economic factors. Previous research highlights differences between Muslim and Christian Arab women regarding breastfeeding initiation and continuation, often influenced by their religiosity and social practices.

The current study aimed to explore the diverse factors contributing to breastfeeding cessation among Arab women in Israel, with a particular focus on how cultural and religious beliefs influence their decision-making. The study investigated socio-demographic characteristics such as age, education, employment, and income, as well as psychological and cultural determinants. Additional factors included the level of breastfeeding guidance received and infant-related health and feeding behaviors. The research sought to provide a comprehensive understanding of the systemic and personal factors that lead to the early cessation of breastfeeding, with a particular focus on how religiosity, family support, and socio-economic factors impact breastfeeding decisions.

The study hypothesized that multiple factors influence the decision to discontinue breastfeeding among Arab women in Israel. These factors include age, ethnic group, level of religiosity, number of children, education, employment status, household income, and guidance received regarding breastfeeding. It was also hypothesized that women who are older, more religious, have higher education levels, or receive better breastfeeding support would be more likely to continue breastfeeding. The study brought to light information that is significant for informing future strategies.

Christian Arab women were found to be more likely to stop breastfeeding earlier compared to their Muslim counterparts. Higher levels of religiosity were associated with an increased likelihood of breastfeeding cessation, contrary to the hypothesis that religiosity would promote continued breastfeeding. This finding suggests that religiosity may play a complex role in breastfeeding decisions, potentially reflecting cultural expectations or religious practices [61].

This association between higher religiosity and increased breastfeeding cessation observed in our study highlights a complex relationship that requires further contextualization. While it may seem counterintuitive, it can be explained by considering several socio-cultural factors that interact with religiosity. In more religious households, there may be greater emphasis on traditional gender roles and larger family sizes, which can increase caregiving demands and reduce the feasibility of sustained breastfeeding. Furthermore, religious norms surrounding modesty and public breastfeeding may discourage women from breastfeeding in surroundings outside the home, leading to the necessity to introduce bottle feeding. Additionally, previous studies suggest that religiosity may influence maternal decision-making, with mothers relying more on family traditions and religious advice rather than professional breastfeeding guidance. Thus, religiosity may interact with a broader set of cultural and social dynamics, rather than acting as a direct determinant of breastfeeding cessation.

While cultural and religious norms in Arab society generally encourage breastfeeding, specific societal expectations and stigmas may create barriers to sustained breastfeeding. Additionally, societal pressures on mothers to balance household responsibilities, care for multiple children, or return to work early can contribute to early breastfeeding cessation. Moreover, some women may feel pressure from family members, including older generations, to introduce complementary feeding early, based on traditional practices rather than professional medical advice.

Moreover, mothers with more children, particularly those with four or more, were more likely to stop breastfeeding earlier. This is likely due to the increased demands of caring for multiple children, which can make breastfeeding more challenging.

Additionally, employment status significantly impacted breastfeeding continuation, with full-time employees more likely to stop breastfeeding due to the challenges of balancing work and breastfeeding. Mothers with heavier workloads were also less likely to continue breastfeeding [62]. In Israel, a survey showed that the breastfeeding initiation rate is around 90%, with all women who began breastfeeding in the hospital reporting exclusive breastfeeding. However, exclusive breastfeeding rates drop significantly during maternity leave, largely due to challenges such as milk supply issues and technical difficulties with breastfeeding [63].

Also, surprisingly, personal guidance received during prenatal care or in the hospital was associated with an increased likelihood of breastfeeding cessation, possibly indicating that the quality of guidance or the circumstances in which it is given may not be fully supportive of sustained breastfeeding [58]. Previous studies have shown that mothers often receive conflicting information from different healthcare providers, which can create confusion and reduce trust in professional guidance. Additionally, the cultural competence of healthcare providers is crucial when addressing the unique needs and values of Arab women in Israel. Culturally sensitive interventions that respect religious and social norms may be more effective in promoting sustained breastfeeding practices, for example, understanding community-specific stigmas, such as discomfort with public breastfeeding.

Furthermore, family members and friends played a significant role in influencing the decision to stop breastfeeding. Financial considerations also contributed to breastfeeding cessation, particularly for women in lower-income households [64].

Lastly, mothers who supplemented breastfeeding with formula or who relied entirely on formula feeding were significantly more likely to stop breastfeeding, highlighting the role of infant feeding behaviors in shaping breastfeeding practices.

One caveat must be made, namely, that our study design is cross-sectional, and as such, the findings reflect associations rather than causation. For example, while our analysis identified a significant relationship between employment status and breastfeeding cessation, we cannot conclude that employment directly causes breastfeeding cessation. Other unmeasured factors, such as workplace policies, family dynamics, or socio-economic pressures may contribute to this relationship. Furthermore, the directionality of this association cannot be determined within the scope of this study. Future longitudinal research is needed to explore the causal mechanisms underlying this relationship, such as whether returning to work leads to early cessation or whether mothers who plan to stop breastfeeding are more likely to return to work earlier.

These findings underscore the complexity of factors that contribute to breastfeeding cessation among Arab women in Israel. The interaction between socio-demographic, religious, and personal factors suggests that tailored interventions are needed to address the specific barriers faced by different subgroups within the population. Understanding these dynamics can inform the development of policies and healthcare practices to promote sustained breastfeeding and improve maternal and infant health outcomes in this population.

## 5. Limitations

It is important to note the limitations of the current research. The sample size, for instance, while sufficient for the analysis, consisted of 349 Arab women from Israel, primarily Muslims (65%) and Christians (35%). Although this sample provides valuable insights into the breastfeeding behaviors of these groups, the findings may not be fully generalizable to all Arab women in Israel or other regions. Cultural diversity within the Arab population and the influence of local socio-economic conditions may limit the applicability of these results to other contexts.

In addition, the reliance on self-reported data introduces potential biases such as recall bias, social desirability bias, and subjective interpretation of questions. Mothers may have underreported or overreported their breastfeeding practices or the factors influencing their decisions based on their perceptions or desire to present themselves in a positive light.

It is important to note that our study design is cross-sectional, and as such, the findings reflect associations rather than causation. For example, while our analysis identified a significant relationship between employment status and breastfeeding cessation, we cannot conclude that employment directly causes breastfeeding cessation. It is possible that other unmeasured factors, such as workplace policies, family dynamics, or socio-economic pressures, may contribute to this relationship. Furthermore, the directionality of this association cannot be determined within the scope of this study. Future longitudinal research is needed to explore the causal mechanisms underlying this relationship, such as whether returning to work leads to early cessation or whether mothers who plan to stop breastfeeding are more likely to return to work earlier.

Moreover, while religiosity was included as a variable, the study measured it on a 5-point 3-item Likert scale, which may not capture the full complexity and diversity of religious beliefs and practices. The influence of specific religious teachings, cultural interpretations, and individual faith experiences could vary widely, making it difficult to fully understand how religiosity impacts breastfeeding behaviors.

Furthermore, while the study sheds light on the breastfeeding practices of Arab women in Israel, it does not allow for comparison with other ethnic groups within the country, such as Jewish or Druze women. Including these groups could have provided more comprehensive insights into the broader social and cultural factors influencing breastfeeding in Israel.

The study primarily relied on quantitative methods and did not explore the qualitative aspects of breastfeeding cessation in-depth. A mixed-methods approach, incorporating interviews or focus groups, could have provided a richer, deeper appreciation of the personal and emotional experiences of mothers and their decision-making processes.

Finally, one limitation of this study is the handling of missing data through listwise deletion, which may have reduced the sample size for certain analyses. Although sensitivity analyses indicated no significant impact on the results, future studies may benefit from using more sophisticated imputation techniques to address missing data.

These limitations suggest that while the study provides valuable insights into the factors influencing breastfeeding cessation among Arab women in Israel, further research is needed to address these constraints and explore additional dimensions of the topic.

## 6. Recommendations

In light of the limitations and the findings emanating from the current study, we posit several suggestions for future research. For example, longitudinal designs need to be considered in order to track breastfeeding practices over time. This approach would allow researchers to explore how factors influencing breastfeeding decisions change as infants grow and as mothers’ circumstances, such as employment or family dynamics, evolve. It would also help establish causality and, thus, offer a more effective direction for encouraging long-term breastfeeding practice.

Additionally, to gain a deeper understanding of the personal and emotional experiences behind breastfeeding cessation, future studies should incorporate qualitative methods such as interviews, focus groups, or case studies. This would allow researchers to explore mothers’ personal narratives, cultural values, religious beliefs, and social pressures that quantitative methods may not capture fully.

For generalizability, expanding the sample size to include more diverse groups, such as Jewish, Druze, and other ethnic or religious populations in Israel, could provide a more comprehensive view of breastfeeding practices across different communities. Additionally, including women from a wider range of socio-economic backgrounds would help clarify how financial factors intersect with cultural and religious beliefs to influence breastfeeding decisions.

Moreover, future research should further investigate the role of healthcare professionals in influencing breastfeeding continuation. This could include an examination of the quality and consistency of breastfeeding guidance provided, as well as an exploration of healthcare providers’ attitudes and knowledge about breastfeeding support. Identifying gaps in the training or resources available to healthcare workers could inform better interventions. Healthcare providers should receive training on delivering consistent and culturally sensitive breastfeeding guidance. This includes understanding the specific cultural and religious values of Arab women and addressing potential stigmas or misconceptions that may hinder breastfeeding continuation.

Also, further studies should explore the complexities of religiosity and cultural beliefs in greater depth, possibly by using more nuanced measures of religiosity that take into account both personal faith practices and broader community norms. Understanding the specific religious teachings and interpretations that influence breastfeeding decisions could help create more targeted and culturally appropriate interventions.

At the workplace, given the significant role of employment in breastfeeding cessation, future research should focus on workplace policies and support systems for breastfeeding mothers. This could include examining the availability and effectiveness of maternity leave, breastfeeding-friendly work environments, and policies allowing for breast milk expression. Identifying best practices in workplace accommodations could inform policies aimed at supporting working mothers. Furthermore, future research should test the effectiveness of interventions designed to promote sustained breastfeeding. These could include educational programs, peer support initiatives, or workplace accommodations aimed at reducing the barriers to breastfeeding. Researchers could measure outcomes such as breastfeeding duration, maternal satisfaction, and infant health to assess the impact of such interventions.

Lastly, future research should consider exploring the relationship between maternal mental health, such as postpartum depression or anxiety, and breastfeeding practices. Understanding how mental health affects breastfeeding continuation could provide valuable insights for developing holistic support systems for mothers. Healthcare interventions aimed at promoting sustained breastfeeding should consider the mental health of mothers and the role of family support dynamics. Providing mental health support during the postpartum period and fostering family involvement in breastfeeding practices could positively influence breastfeeding continuation.

By addressing these areas, future research can contribute to a more complete understanding of the complex factors influencing breastfeeding decisions and help shape policies and interventions that promote sustained breastfeeding among diverse populations.

## 7. Conclusions

The study investigates the factors contributing to breastfeeding cessation among Arab women in Israel, focusing on the socio-demographic, cultural, religious, and personal determinants that influence their decisions to stop breastfeeding. Despite the known health benefits of breastfeeding, many women discontinue breastfeeding earlier than recommended. This study explores how cultural norms, religious beliefs, maternal education, employment status, and family support shape breastfeeding practices among Israeli Arab women, particularly in Muslim and Christian communities.

The findings highlight significant differences between Muslim and Christian Arab women, with Christian women being more likely to stop breastfeeding earlier. Factors such as the number of children, religiosity, employment status, and the nature of breastfeeding guidance received also play important roles. For example, mothers with more children or heavier workloads tend to cease breastfeeding sooner, while those with supportive family networks are more likely to continue breastfeeding. Surprisingly, personal guidance was linked to an increased likelihood of breastfeeding cessation, suggesting that the quality of professional support may not be fully adequate.

The study also underscores the significant role of infant-related factors, such as reliance on formula feeding, which contributes to early breastfeeding cessation. Family influence, financial pressures, and the baby’s health also play crucial roles in shaping breastfeeding decisions.

The study acknowledges several limitations, including the reliance on self-reported data, the cross-sectional design, and a sample that may not fully represent all Arab women in Israel. It calls for future longitudinal and qualitative research to provide deeper insights and to explore how workplace policies and healthcare practices can better support breastfeeding mothers.

In conclusion, the study provides valuable insights into the complex factors that lead to breastfeeding cessation among Arab women in Israel, offering recommendations for more tailored interventions that could promote sustained breastfeeding and improve maternal and infant health outcomes in this population.

## Figures and Tables

**Table 1 nutrients-17-00735-t001:** Logistic regression coefficients for reason #1 in relation to stopping breastfeeding.

Predictor	*b*	*SE*	*Sig.*	Exp(B)	LL_95%_	UL_95%_
Ethnicity (0 = Muslim; 1 = Christian)	0.95	0.34	0.005	2.58	1.33	4.99
Number of Children (ref = none or 1 child)			0.001			
2–3 Children	0.51	0.35	0.149	1.66	0.83	3.32
4 or more Children	1.64	0.45	0.000	5.18	2.13	12.60
Job Scope (0% = unemployed; 100% = full-time)	−1.16	0.38	0.002	0.31	0.15	0.66
Breastfeeding Guidance (ref = did not receive guidance)			0.011			
Group Guidance	−0.01	0.43	0.974	0.99	0.42	2.31
Personal Guidance	0.89	0.34	0.008	2.42	1.26	4.68
Baby’s Nutrient Source (ref = exclusive breastfeeding)			0.000			
Baby Food/Formula Only	3.43	0.50	0.000	31.03	11.53	83.48
Combined Breastfeeding	1.54	0.38	0.000	4.64	2.21	9.75
Family Members and/or Friends	0.65	0.21	0.002	1.91	1.27	2.87
Financial Considerations	0.59	0.20	0.004	1.80	1.21	2.69

Notes. *b* = unstandardized regression coefficient (i.e., slope). *SE* = standard error. *Sig*. = exact significance level. Exp(B) = log-likelihood ratios. LL and UL = lower and upper limits, respectively, of 95% confidence interval. ref = reference category (only for categorical variables).

**Table 2 nutrients-17-00735-t002:** Logistic regression coefficients for reason #2 in relation to stopping breastfeeding.

Predictor	*b*	*SE*	*Sig.*	Exp(B)	LL_95%_	UL_95%_
Ethnicity (0 = Muslim; 1 = Christian)	2.56	0.59	0.000	12.89	4.03	41.22
Religiosity	1.21	0.51	0.017	3.37	1.24	9.13
Number of Children (ref = none or 1 child)			0.002			
2–3 Children	−0.17	0.68	0.803	0.84	0.22	3.21
4 or more Children	1.84	0.74	0.013	6.31	1.47	27.06
Education (0 = non-academic; 1 = academic)	1.47	0.59	0.013	4.35	1.36	13.91
Breastfeeding Duration (ref = did not breastfeed at all)			0.012			
1–4 Months	5.27	2.27	0.020	194.81	2.26	16791.48
More than 4 Months	3.52	2.14	0.101	33.63	0.50	2243.44
Baby’s Age (months) (ref = 4–5 months)			0.000			
5–6 Months	2.16	0.96	0.025	8.64	1.32	56.70
More than 6 Months	−1.78	0.69	0.009	0.17	0.04	0.65
Family Members and/or Friends	0.99	0.37	0.007	2.70	1.32	5.54
Financial Considerations	1.49	0.31	0.000	4.43	2.43	8.08
Information (from literature, media, internet)	−1.19	0.26	0.000	0.30	0.18	0.50

Notes. *b* = unstandardized regression coefficient (i.e., slope). *SE* = standard error. *Sig*. = exact significance level. Exp(B) = log-likelihood ratios. LL and UL = lower and upper limits, respectively, of 95% confidence interval. ref = reference category (only for categorical variables).

**Table 3 nutrients-17-00735-t003:** Logistic regression coefficients for reason #3 in relation to stopping breastfeeding.

Predictor	*b*	*SE*	*Sig.*	Exp(B)	LL_95%_	UL_95%_
Religiosity	0.94	0.42	0.026	2.55	1.12	5.81
Employment Status (ref = unemployed)			0.012			
Self-employed	1.91	0.65	0.003	6.77	1.91	24.02
Part- or Full-time Employee	0.39	0.38	0.293	1.48	0.71	3.10
Financial Considerations	1.21	0.22	0.000	3.34	2.19	5.09
Information (from literature, media, internet)	−0.52	0.15	0.000	0.59	0.44	0.79

Notes. *b* = unstandardized regression coefficient (i.e., slope). *SE* = standard error. *Sig*. = exact significance level. Exp(B) = log-likelihood ratios. LL and UL = lower and upper limits, respectively, of 95% confidence interval. ref = reference category (only for categorical variables).

**Table 4 nutrients-17-00735-t004:** Logistic regression coefficients for reason #4 in relation to stopping breastfeeding.

Predictor	*b*	*SE*	*Sig.*	Exp(B)	LL_95%_	UL_95%_
Religiosity	−0.42	0.21	0.042	0.65	0.43	0.99
Number of Children (ref = none or 1 child)			0.003			
2–3 Children	−0.06	0.40	0.875	0.94	0.43	2.05
4 or more Children	1.28	0.45	0.005	3.59	1.48	8.69
Breastfeeding Guidance (ref = did not receive any guidance)			0.001			
Group Guidance	1.18	0.50	0.018	3.25	1.22	8.65
Personal Guidance	1.49	0.41	0.000	4.44	1.98	9.97
Baby’s Nutrient Source (ref = exclusive breastfeeding)			0.000			
Baby Food/Formula Only	2.01	0.43	0.000	7.43	3.19	17.30
Combined Breastfeeding	0.51	0.42	0.223	1.66	0.73	3.75

Notes. *b* = unstandardized regression coefficient (i.e., slope). *SE* = standard error. *Sig*. = exact significance level. Exp(B) = log-likelihood ratios. LL and UL = lower and upper limits, respectively, of 95% confidence interval. ref = reference category (only for categorical variables).

**Table 5 nutrients-17-00735-t005:** Logistic regression coefficients for reason #5 in relation to stopping breastfeeding.

Predictor	*b*	*SE*	*Sig.*	Exp(B)	LL_95%_	UL_95%_
Exclusive Breastfeeding Duration (ref = did not breastfeed)			0.021			
1–4 Months	−0.28	0.67	0.682	0.76	0.20	2.84
More than 4 Months	−1.21	0.63	0.049	0.30	1.02	1.63
Family Members and/or Friends	0.73	0.23	0.002	2.07	1.31	3.26

Notes. *b* = unstandardized regression coefficient (i.e., slope). *SE* = standard error. *Sig*. = exact significance level. Exp(B) = log-likelihood ratios. LL and UL = lower and upper limits, respectively, of 95% confidence interval. ref = reference category (only for categorical variables).

**Table 6 nutrients-17-00735-t006:** Logistic regression coefficients for reason #6 in relation to stopping breastfeeding.

Predictor	*b*	*SE*	*Sig.*	Exp(B)	LL_95%_	UL_95%_
Number of Children (ref = none or 1 child)			0.034			
2–3 Children	0.19	0.34	0.582	1.21	0.62	2.35
4 or more Children	0.99	0.40	0.013	2.70	1.23	5.91
Job Scope (0% = unemployed; 100% = full-time)	1.26	0.40	0.002	3.51	1.61	7.65
Currently Breastfeeding (0 = no; 1 = yes)	−0.84	0.32	0.008	0.43	0.23	0.80
Good Bond/Attachment with the Baby	1.38	0.38	0.000	3.96	1.87	8.38

Notes. *b* = unstandardized regression coefficient (i.e., slope). *SE* = standard error. *Sig* = exact significance level. Exp(B) = log-likelihood ratios. LL and UL = lower and upper limits, respectively, of 95% confidence interval. ref = reference category (only for categorical variables).

## Data Availability

Individual level data cannot be made publicly available due to legal and ethical restrictions. Aggregative data might be provided upon reasonable request to the corresponding author.

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
