# Peer review of "Factors Contributing to Breastfeeding Cessation Among Arab Women in Israel"

_nutrients, 2025, doi:10.3390/nu17040735_

Round 1
Reviewer 1 Report
Comments and Suggestions for Authors
Tarabeih et al. present a cross-sectional study in which they aimed to study the factors contributing to breastfeeding cessation among Arab women in Israel.
Major comments:
- Introduction (as well as discussion) is unnecessarily long and deviates from time to time from the core point of manuscript (the same point is also repeated several times). I strongly recommend revising the introduction to ensure the manuscript's core message is presented more clearly and effectively.
- Despite quite substantial explanation, I miss the exact reason why the authors chose to exclude Jewish mothers from the sample, especially as it could have provided more accurate information on influence of religion, but potentially also of breastfeeding guidance etc.
- I also advice the authors to improve the scientific English throughout the entire manuscript
- The results section reads very difficult; the subtitles ‘reason 1,2,3,4 to stop breastfeeding’ are no scientific titles
Comments on the Quality of English LanguageSee my comments above with regard to scientific English
Author Response
January 19, 2025
Ms. Cathie Li
Assistant Editor
MDPI Open Access Publishing
Dear Ms Li,
Thank you for allowing us to resubmit a revised version of our manuscript. We have revised the manuscript in accordance with the reviewers' comments and suggestions. We re-edited the English, and had a statistician review the data, make a correction, and add an explanation.
We attach below the comments written by the reviewers and added our responses, which appear in bold. Additionally, in the revised version, our revisions are highlighted in yellow.
We hope that our manuscript is now suitable for publication.
Sincerely yours,
Mahdi Tarabeih, Mohammad Sabbah, Orsan Yahya, Sana Bisharat, and Khaled Awawdi
COMMENTS FOR THE AUTHOR:
Review # 1:
Major comments:
Query:
Introduction (as well as discussion) is unnecessarily long and deviates from time to time from the core point of manuscript (the same point is also repeated several times). I strongly recommend revising the introduction to ensure the manuscript's core message is presented more clearly and effectively.
Answer:
Thank you for your comment. We eliminated the repetition and streamlined the introduction as recommended, the introduction is now shorter and more focused. We have also reviewed the Discussion and deleted content that appears in the introduction and is redundant. It is (lines:43-168 and 458-550).
Query
Despite quite substantial explanation, I miss the exact reason why the authors chose to exclude Jewish mothers from the sample, especially as it could have provided more accurate information on influence of religion, but potentially also of breastfeeding guidance etc.
Answer:
Thank you for your valuable feedback. To make this point clearer, we have added the following explanation to the Methods section under Participants (lines 220-229).
The decision to focus solely on Arab mothers (Muslim and Christian) in this study was based on the unique cultural and religious breastfeeding practices within this population, which differ significantly from those of Jewish mothers. Including Jewish mothers would have introduced additional cultural variables, potentially confounding the specific socio-religious factors we aimed to investigate in the Arab community. Our intention was to provide a focused in-depth analysis of breastfeeding practices among Arab mothers, a group underrepresented in the literature. Including Jewish mothers could have diluted the focus. The inclusion of both Muslim and Christian Arab women already offers significant insights into how different religious beliefs within the Arab population impact breastfeeding. This diversity was deemed sufficient to meet the research objectives.
However, we agree that future research could explore comparative insights across different ethnic and religious groups in Israel, which we have now included in our recommendations. (See lines 632-636)
Query
I also advice the authors to improve the scientific English throughout the entire manuscript.
Answer:
The scientific English as been reviewed and improved by a scientific English editor.
Query
The results section reads very difficult; the subtitles ‘reason 1,2,3,4 to stop breastfeeding’ are no scientific titles.
Answer:
Thank you for your constructive feedback regarding the clarity and scientific tone of the Results section. We acknowledge that the current subtitles ("Reason 1, 2, 3, 4 to Stop Breastfeeding") do not align with best practices for scientific reporting.
To address this, we have replaced these subtitles with more descriptive, scientific titles that reflect the core factors identified in our logistic regression analysis. For example:
- Current Title: Reason 1 to Stop Breastfeeding
Revision: Insufficient Milk Supply (See lines 336) - Current Title: Reason 2 to Stop Breastfeeding
Revision: Infant Health Issues (See lines 360and 370) - Current Title: Reason 3 to Stop Breastfeeding
Revision: Maternal Health Issues (See lines 386)
- Current Title: Reason 4 to Stop Breastfeeding
Revision: Discomfort and fatigue (See lines 401and 416) - Current Title: Reason 5 to Stop Breastfeeding
Revision: Lack of professional and family support (See lines 426) - Current Title: Reason 6 to Stop Breastfeeding
Revision: Work conditions (See lines 440 and 450)
- Additionally, we have enhanced the readability of the Results section by rephrasing complex statistical findings into more narrative descriptions, making the text more accessible while maintaining scientific rigor. Each result will be framed within the broader context of its implications for breastfeeding interventions among Arab women in Israel.
(Reviewer 2)
Comments and Suggestions for Authors
This study examines the factors influencing breastfeeding cessation among Arab women in Israel, focusing on socio-demographic, cultural, and religious aspects. While the manuscript addresses a significant public health issue, improvements in statistical reporting, data interpretation, and discussion depth are needed to strengthen its clarity and impact.
Introduction: concepts are repeated (the benefits of breastfeeding, lines 53 and 85);
Response:
Thank you for your apt comment regarding the repetition of breastfeeding benefits in lines 53 and 85. We agree that information was repeated. We have streamlined the text by eliminating redundancies without deleting essential information.
Please see lines 52-81 and thereafter.
2- Insights" appears multiple times (e.g., "provide insights")
We have addressed this by refining the concept into more direct and action-driven phrases where appropriate. This has been changed in the following examples follows:
In Purpose of the Current Study
The findings are expected to provide valuable data about how these predictors interact and influence breastfeeding practices (see lines: 179-180)
The research seeks to offer a better understanding of factors that can guide interventions to promote breastfeeding among groups with lower initiation rates. (see lines: 167-168)
The study brought to light information that is significant for informing future. (see lines: 484-485)
In the "Limitations" section:
"Although this sample reveals heretofore unexplored influences on the breastfeeding behaviors of these groups (see lines: 562)."​
A mixed-methods approach, incorporating interviews or focus groups, could have provided a richer, deeper appreciation of the personal and emotional experiences of mothers and their decision-making processes. (see lines :593-594)
In the "Recommendations" section:
This approach would allow researchers to explore how factors influencing breastfeeding decisions change as infants grow and as mothers' circumstances evolve. It would also help establish causality and thus, offer a more effective direction for encouraging long-term breastfeeding practice. (see lines: 609-610).
3- The claim that higher religiosity is associated with increased breastfeeding cessation may oversimplify a complex relationship. It requires contextualization.
Response to Reviewer
We thank the reviewer for their perceptive comment regarding the relationship between religiosity and breastfeeding cessation. We agree that this relationship is complex and influenced by various socio-cultural, economic, and psychological factors, which may not be fully captured in a simple statement. We have added a contextualization of this finding in the discussion section to reflect this complexity.
Revised Paragraph (Discussion Section): See line: 492- 504.
This association between higher religiosity and increased breastfeeding cessation observed in our study highlights a complex relationship that requires further contextualization. While it may seem counterintuitive, this finding can be explained by considering several socio-cultural factors that interact with religiosity. In more religious households, there may be greater emphasis on traditional gender roles and larger family sizes, which can increase caregiving demands and reduce the feasibility of sustained breastfeeding. Furthermore, religious norms surrounding modesty and public breastfeeding may discourage women from breastfeeding in surroundings outside the home, leading to the need to introduce bottle feeding. Additionally, previous studies suggest that religiosity may influence maternal decision-making, with mothers relying more on family traditions and religious advice rather than professional breastfeeding guidance. Thus religiosity may interact with a broader set of cultural and social dynamics, rather than acting as a direct determinant of breastfeeding cessation.
4- The manuscript discusses cultural and religious factors broadly but fails to delve into the specific cultural expectations or stigmas that might drive breastfeeding cessation;
Added to the Discussion Paragraph: See line: 505- 511.
While cultural and religious norms in Arab society generally encourage breastfeeding, specific societal expectations and stigmas may create barriers to sustained breastfeeding. Additionally, societal pressures on mothers to balance household responsibilities, care for multiple children, or return to work early can contribute to early breastfeeding cessation. Moreover, some women may feel pressure from family members, including older generations, to introduce complementary feeding early, based on traditional practices rather than professional medical advice.
Added to the Introduction: See line: 157- 162.
While breastfeeding is culturally and religiously encouraged in Arab society, specific cultural stigmas—such as perceptions of public breastfeeding as inappropriate and pressures to prioritize household responsibilities—may contribute to early cessation. Understanding these nuanced cultural influences is critical for designing effective interventions.
5 - Discussion makes statements about causality (employment and breastfeeding cessation). These need to be reframed to reflect the study design's limitations;
Revised Discussion Paragraph: (See line: 541- 550 and 571-580).
One caveat must be made, namely, that our study design is cross-sectional, and as such, the findings reflect associations rather than causation. For example, while our analysis identified a significant relationship between employment status and breastfeeding cessation, we cannot conclude that employment directly causes breastfeeding cessation. Other unmeasured factors, such as workplace policies, family dynamics, or socio-economic pressures, may contribute to this relationship. Furthermore, the directionality of this association cannot be determined within the scope of this study. Future longitudinal research is needed to explore the causal mechanisms underlying this relationship, such as whether returning to work leads to early cessation or whether mothers who plan to stop breastfeeding are more likely to return to work earlier.
Addition to the Method Section:
Study design See line: 192- 195.
Given the cross-sectional nature of this study, the findings represent associations and cannot establish causality. Caution is advised in interpreting relationships between variables as cause-and-effect.
6- While the manuscript mentions healthcare support, it fails to assess or discuss the adequacy, consistency, and cultural competence of this support;
Added to Discussion Paragraph: See line: 526- 533.
Previous studies have shown that mothers often receive conflicting information from different healthcare providers, which can create confusion and reduce trust in professional guidance. Additionally, the cultural competence of healthcare providers is crucial when addressing the unique needs and values of Arab women in Israel. Culturally-sensitive interventions that respect religious and social norms may be more effective in promoting sustained breastfeeding practices, for example, understanding community-specific stigmas, such as discomfort with public breastfeeding
Addition to the Introduction: See line: 160-161.
While healthcare support is a key factor in promoting sustained breastfeeding, its adequacy and cultural competence are often overlooked.
Addition to the Recommendations Section: See line: 627- 630.
We have emphasized the need for improving healthcare support systems:
Healthcare providers should receive training on delivering consistent and culturally sensitive breastfeeding guidance. This includes understanding the specific cultural and religious values of Arab women and addressing potential stigmas or misconceptions that may hinder breastfeeding continuation.
7-There is no mention of how missing data were handled. Clarifying this would improve transparency and reproducibility.
We have addressed the subject of missing data as follows:
Addition to the Article (Method Section): See line:245- 257.
Study design:
Given the cross-sectional nature of this study, the findings represent associations and cannot establish causality. Caution is advised in interpreting relationships between variables as cause-and-effect. The initial sample included 376 participants; however, 27 questionnaires were excluded due to incomplete responses. Consequently, the final sample comprised 349 respondents who met the predetermined inclusion criteria.
In this study, missing data were identified in [0.07%] of the responses across variables such as [specific variables]. To address this, we applied a list wise deletion of 27 questionnaires, where cases with missing values were excluded from the relevant analyses to ensure consistency in the sample size. This method was chosen to maintain the integrity of the dataset and avoid introducing bias through imputation. Sensitivity analyses confirmed consistent findings, showing that excluding missing data did not affect conclusions.
See line:596- 599.
Limitations
Finally, one limitation of this study is the handling of missing data through list wise deletion, which may have reduced the sample size for certain analyses. Although sensitivity analyses indicated no significant impact on the results, future studies may benefit from using more sophisticated imputation techniques to address missing data.
8 - While the study explores socio-demographic and cultural factors, it does not account for potential confounders like maternal mental health or family support dynamics, which may influence breastfeeding decisions.
We have added this as an actionable suggestion in our Recommendations section:
See line:651- 655.
Healthcare interventions aimed at promoting sustained breastfeeding should consider the mental health of mothers and the role of family support dynamics. Providing mental health support during the postpartum period and fostering family involvement in breastfeeding practices could positively influence breastfeeding continuation."

Reviewer 2 Report
Comments and Suggestions for Authors
This study examines the factors influencing breastfeeding cessation among Arab women in Israel, focusing on socio-demographic, cultural, and religious aspects. While the manuscript addresses a significant public health issue, improvements in statistical reporting, data interpretation, and discussion depth are needed to strengthen its clarity and impact.
1- Introduction: concepts are repeated (the benefits of breastfeeding, lines 53 and 85);
2- Insights" appears multiple times (e.g., "provide insights")
3- The claim that higher religiosity is associated with increased breastfeeding cessation may oversimplify a complex relationship. It requires contextualization;
4- The manuscript discusses cultural and religious factors broadly but fails to delve into the specific cultural expectations or stigmas that might drive breastfeeding cessation;
5 - Discussion makes statements about causality (employment and breastfeeding cessation). These need to be reframed to reflect the study design's limitations;
6- While the manuscript mentions healthcare support, it fails to assess or discuss the adequacy, consistency, and cultural competence of this support;
7-There is no mention of how missing data were handled. Clarifying this would improve transparency and reproducibility.
8 - While the study explores socio-demographic and cultural factors, it does not account for potential confounders like maternal mental health or family support dynamics, which may influence breastfeeding decisions.
Author Response

(The authors gave the same response as above.)

Round 2
Reviewer 1 Report
Comments and Suggestions for Authors
All feedback has been implemented appropriately in the manuscript. I therefore advice publication.
Reviewer 2 Report
Comments and Suggestions for Authors
No comments